

# Suppression of very low frequency radio noise in transient electromagnetic data with semi-tapered gates

Jakob Juul Larsen[1,5], Stine Søgaard Pedersen[2,3], Nikolaj Foged[2,5], and Esben Auken[4]

[1]Department of Engineering, Aarhus University, Finlandsgade 22, 8200 Aarhus N, Denmark
[2]Hydrogeophysics Group, Department of Geoscience, Aarhus University, C. F. Møllers Alle 4, 8000 Aarhus C, Denmark
[3]Department of Physics and Astronomy, Aarhus University, Ny Munkegade 120, 8000 Aarhus C, Denmark
[4]Geological Survey of Denmark and Greenland, Øster Voldgade 10, 1350 Copenhagen, Denmark
[5]WATEC, Aarhus University Centre for Water Technology, Ny Munkegade 120, 8000 Aarhus C, Denmark

**Correspondence:** Jakob Juul Larsen (jjl@eng.au.dk)

**Abstract.** The transient electromagnetic method (TEM) is widely used for mapping subsurface resistivity structures but data are inevitably contaminated by noise from various sources. It is common practice to gate signals from TEM systems to reduce the amount of data and improve the signal-to-noise ratio (SNR). Gating acts as a filter, and optimum gating will pass the TEM signal un-attenuated while suppressing noise. In systems based on analog boxcar integrators, the gating corresponds to filtering

with a square window. The frequency response of this window shape has large side lobes, which are often insufficient in attenuating noise, e.g., from radio signals in the very low frequency (VLF) 3-30 kHz band. Tapered gates have better side lobe suppression and attenuates noise better but tapering with analog boxcar integrators is difficult. We propose to use many short boxcar gates, denoted sub-gates, and combine the sub-gates into semi-tapered gates to improve noise rejection at late gates where low signal normally leads to poor SNR. The semi-tapering approach is analyzed and tested experimentally on data from

a roving TEM system. We quantify the effect of semi-tapered gates by computing an improvement factor as the ratio between the standard error of data measured with boxcar gates and the standard error of data measured with semi-tapered gates. Data from a test survey in Gedved, Denmark with 1825 measurements gave mean improvement factors between 1.04 and 2.22 for the ten late-time gates centered between 78.7 $\mu$s and 978.1 $\mu$s. After inversion of the data, we find that semi-tapering increases the depth of investigation by about 20 % for this specific survey. We conclude that the semi-tapered approach is a viable path

towards increasing SNR in TEM systems based on analog boxcar integrators.

## 1  Introduction

The transient electromagnetic method (TEM) is a widely applied geophysical method for delineating resistivity and resistivity structures in the subsurface of the earth. The method has found extensive use in many areas including mineral exploration,

groundwater mapping, and geotechnical surveys, see Auken et al. (2017) for a recent review. Importantly, TEM instruments





can be mounted on roving systems, which allows for high-resolution and cost-efficient mapping of large areas. Roving TEM systems include both fixed wing and helicopter airborne systems, as well as ground-based systems (e.g., Balch et al., 2003; Mulé et al., 2012; Auken et al., 2019).

The principle behind TEM is that a current applied to a transmitter coil generates a primary magnetic field. When the primary field is turned off, eddy currents are generated in the earth, which in turn generate a secondary magnetic field. The decay of the secondary field is measured by a receiver, typically an induction coil. The resistivity structure of the sub-surface earth is encoded in the secondary field and can be retrieved through inversion (Nabighian and Macnae, 1991).

Like any other electric or electromagnetic geophysical method, TEM is affected by electromagnetic noise. The noise is comprised of contributions from multiple sources including sinusoidal power grid components at 50 Hz / 60 Hz and harmonics thereof, spherics from thunderstorms, very low frequency (VLF, 3-30 kHz) radio communication signals, internal noise from electronic components as well as motion noise in roving systems (Macnae et al., 1984; Rasmussen et al., 2018a). Signal processing methods have been developed to mitigate the effects of noise. Important examples include suppression of power grid noise through synchronous detection using alternating polarity transmitter pulses, culling of signals affected by spherics, modelling and subtraction of VLF noise, as well as gating and stacking of signals (see e.g. Macnae et al., 1984; Nyboe and Sørensen, 2012; Macnae, 2015; Rasmussen et al., 2018b).

TEM measurements cover a huge dynamic range; signal values can span over six orders of magnitude on a time scale ranging from $\mu s$ to tens of ms. Historically, constraints in available technology implied that it was not possible continuously to sample the decaying TEM signal at the required dynamic range and store these data sets for later use. Instead, a common strategy was to use an analog integrator to boxcar gate the signal in exponentially increasing gate widths. The TEM signal approximately decays as $t^{-5/2}$. The analog integration over time gives an output decaying as $t^{-3/2}$, reducing the need for dynamic range in the subsequent analog-to-digital (A/D) conversion. In addition, the sampling period of the A/D converter can reduced to that of the narrowest gate. In typical applications, 8-10 boxcar gates per time decade are used, enough to provide adequate representation of the TEM signal for inversion yet offering a significant reduction in the amount of data to be stored (Munkholm and Auken, 1996).

The boxcar gating corresponds to filtering of the data with a square window (Harris, 1978). The width of the boxcar gates in the time domain is inversely proportional to the width of the main lobe in the frequency domain. The exponentially increasing gate widths therefore correspond to filters with decreasing main lobe widths. This matches TEM signals, which extends to high frequencies at early times, but has low frequency content only at later times. Unfortunately, the frequency response associated with square windows has large side lobes, implying that noise is not necessarily efficiently suppressed. One particular example of this is VLF noise where different radio transmitters give rise to distinct noise peaks in the frequency spectrum. We note that while only a few papers has dealt with the suppression of VLF noise in TEM measurements, there are vast research fields and associated literature devoted to understanding the properties of VLF signals as well as utilizing the VLF signals in, for example, ionospheric studies and geophysical surveying (see e.g. Barr et al., 2000; Oskooi and Pedersen, 2005; Inan et al., 2010; Eppelbaum and Mishne, 2011).





The purpose of this work is to investigate the properties of an intermediate gating strategy, applicable to TEM systems equipped with analog boxcar integrators. Specifically, we increase the number of boxcar gates per decade and form weighted combinations of these to produce semi-tapered gates with improved noise suppression properties. The reduction of noise, in particular VLF noise in the 3-30 kHz range, improves the data quality and extends the range of usable gates, which in turn leads to an increased depth of investigation.

The outline of the paper is as follows. First, we give a theoretical analysis and comparison of boxcar, tapered, and semi-tapered gates. Second, we present and compare experimental results of TEM data acquired with boxcar and semi-tapered gates.

## 2    Methods

### 2.1    Gating

At early times, the TEM signal extends from DC to high frequencies and is high amplitude. At late times, the TEM signal only extends from DC to low frequencies and is low amplitude. This is illustrated in the double-logarithmic plot in Fig. 1(a) where the TEM signal appears as a line. To improve on the signal-to-noise ratio and reduce the amount of data to be stored, it is customary to integrate the signal in adjacent time windows, called gates. The width of these gates must be chosen so the TEM signal is nearly constant within the gate. This requires short gates at early times where the signal decays fast. At late times, the

signal decays slowly and longer gates can be used. Further, at late times the signal amplitude is reduced and longer gates are beneficial as they integrate the signal over a longer time span and hence improve the SNR. These considerations lead to the common choice of exponentially increasing gate widths.

In systems with a single analog integrator, the gates are necessarily boxcar shaped and non-overlapping, Fig. 1(b). For systems where the TEM signal is continuously digitized at a high sampling rate, any time window, i.e., any gate shape, can

easily be applied in subsequent processing of data and gates can overlap, Fig. 1(c), (Nyboe and Mai, 2017). Figure 1(d) present an intermediate approach where an analog integrator outputs many short boxcar gates, denoted sub-gates in the following. By forming a weighted combination of the sub-gates in subsequent processing, we create semi-tapered gates, with potentially improved noise suppression capabilities similar to continuously digitized sampling.

In essence, gating, i.e., windowing and integration of data acts as a filter with a frequency response given by the Fourier

transform of the gate shape (Harris, 1978). Hence, by applying carefully designed window shapes, selected frequency bands can be suppressed. We emphasize the following general properties about windows and their magnitude frequency response:

–    Increasing the length of a window will decrease the width of the main lobe in the frequency domain.

–    In the case of windows of equal length, the boxcar window has the narrowest main lobe, but also the largest side lobes.

–    Tapered windows have lower amplitude side lobes than boxcar windows.



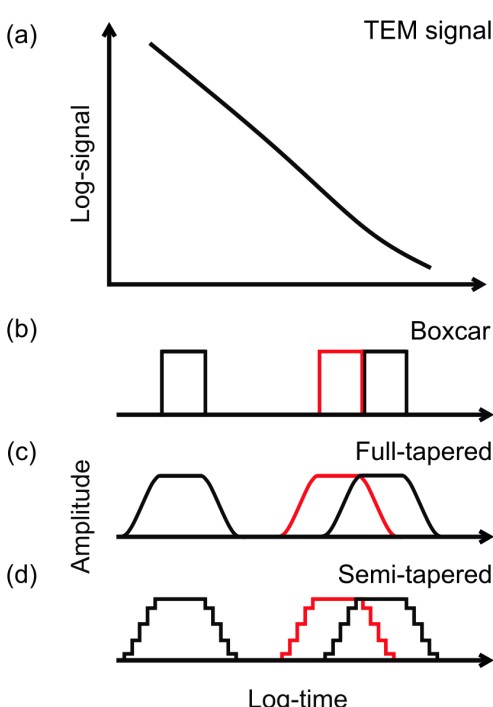

**Figure 1.** The plot in (a) illustrates a decaying TEM signal. The plots in (b), (c), and (d) shows the rectangular, full-tapered and semi-tapered gates, respectively. For simplicity only three gates are shown for each gating strategy and coloured to make them distinguishable. Both full-tapered and semi-tapered gates overlap with neighbouring gates. The gate widths are exponentially increasing, hence gates appear with identical widths on the logarithmic time axis.

Figure 2 illustrates this in detail for a gate centered at 419.7 $\mu$s. In Fig. 2(a) three different gate shapes are plotted: a boxcar gate (full blue line), a full-tapered gate (dashed black line) consisting of a flat main segment identical to the boxcar gate as well as a smooth half-cosine tapering at both edges and a semi-tapered gate (full red line) also consisting of a flat main segment identical to the boxcar gates and non-smooth tapering at both edges. Each taper is made up of five sub-gates with values matching the smooth half-cosine at the sub-gate centers. The half-cosine tapers are symmetric in logarithmic time as shown in

the figure, and therefore non-symmetric in linear time. Corresponding plots of the magnitude frequency response are shown in Fig. 2(b). The responses have been normalized for unit gain at 0 Hz. We note the following two specific features from the magnitude frequency response plot: First, the width of the main lobe is wider for the boxcar gate than for the tapered and semi-tapered gate. This counter-intuitive result is caused by the increased widths of the tapered gates, narrowing the main lobe. For the three gates shown in Fig. 2, the widths, measured as full width at half maximum, are 114.6 $\mu$s (boxcar) and 235.5 $\mu$s (full-

and semi-tapered). Second, both the full-tapered and semi-tapered windows have, on average, almost identical and significantly lower amplitude side lobes, which improves overall noise suppression in the VLF band.



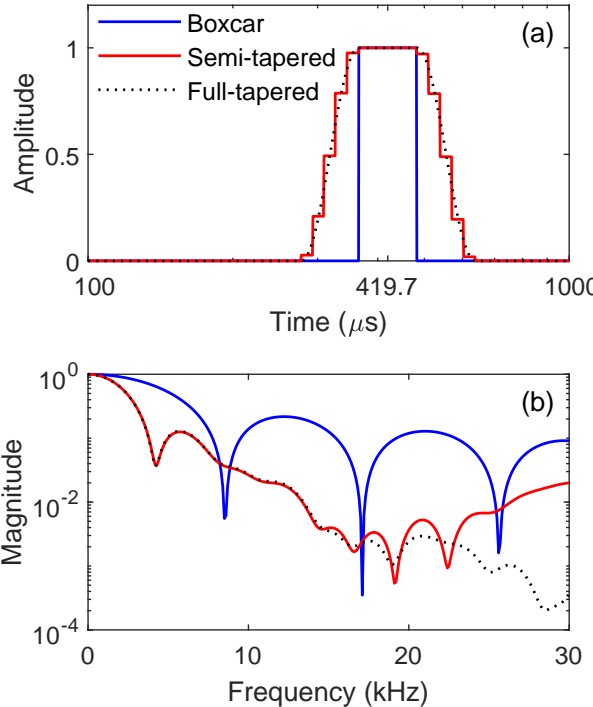

**Figure 2.** Comparison of boxcar, full-tapered and semi-tapered gates in the time domain (a) and frequency domain (b). The gate center time is 419.7 $\mu$s. The boxcar gate width is 114.6 $\mu$s and the semi-tapered width is 235.5 $\mu$s (gate 17 in Table 1).

The half-cosine tapers (full- or semi-) added to the center boxcar gate is only one particular choice of gate shape. We also performed simulations and experiments with other gate shapes, including Gaussian gates and boxcar gates augmented with linear tapers Harris (1978). We found that the performance of boxcar gates augmented with half-cosine tapers gave good results for the tested scenarios, but also that other gate shapes could have better performance in specific scenarios. This is not a surprising result as the effectiveness in noise suppression depends on the exact match between the spectrum of the noise and the frequency response of the gate. For instance, if the frequency of a given radio transmitter coincides with one of the dips in the gate magnitude frequency response, the transmitter will be efficiently suppressed.

## 2.2 The tTEM system

Measurements are conducted with a ground-based towed transient electromagnetic (tTEM) system fully described in Auken et al. (2019). Briefly, the tTEM system is towed behind an all-terrain vehicle and comprises a 2 m by 4 m transmitter coil and a 0.5 m by 0.5 m receiver coil placed 7 m behind the transmitter coil. The system uses dual transmitter moments (low and high) to obtain early and late time TEM data, roughly corresponding to shallow and deep geological layers. In the low moment, the transmitter pulse repetition rate is 2110 Hz and the first usable gate lies at ≈4 $\mu$s after begin of current turn-off. In the high moment, the transmitter pulse repetition rate is 660 Hz, and the first usable gate lies at approximately 9 $\mu$s after current turn-

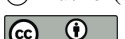



off. High moment measurements last up to 1000 $\mu$s after turn-off with the limit set by the pulse repetition rate. The receiver system contains an analog integrator to boxcar gate the TEM signal. In the standard configuration, a low-moment measurement contains 15 boxcar gates with center times from 4 $\mu$s to 33 $\mu$s and a high-moment measurement contains 23 boxcar gates with center times from 10 $\mu$s to 900 $\mu$s. The output from the analog integrator is sampled with an ADC and stored. The tTEM system

uses synchronous detection with alternating polarity transmitter pulses and the recorded gate values are therefore sign-corrected before stacking (Appendix A). A low moment stack, i.e., one sounding, is made up of 422 individual transients, and one high moment stack is made up of 252 individual transients. In normal operation, the system interleaves between low-moment and high-moment stacks.

The tTEM system can be configured to output up to a maximum of 84 boxcar gates distributed in the low and high moment.

The start time and gate width of each boxcar gate is individually controlled with a 0.65 $\mu$s minimum gate width and a 0.35 $\mu$s boxcar integrator reset time between gates. The limit of 84 gates is imposed by the communication speed of the electronics.

## 3   Experimental Results

A test survey was carried out on farming fields in Gedved, Denmark. The area is characterized by glacial sediment deposits and the resistivity ranges from 8-500 $\Omega m$. The survey site was selected based on the condition that the later gates, i.e. from

approximately 100 $\mu$s and onwards, should be dominated by noise, including VLF radio noise. This ensures that differences in noise rejection between different gating strategies are readily discernible. The driving conditions on the site were good and no motion-induced noise artefacts were observed in the processed data.

For this study, we focus on the high moment data part, where VLF radio noise can be non-negligible at late gates. The tTEM system was configured to record 84 boxcar gates, denoted sub-gates in the following. The center times and widths exponentially

increase within the limitations enforced by the receiver electronics. The first useable high-moment sub-gate is gate 4 centered at 9.2 $\mu$s with a 1.65 $\mu$s width. The last sub-gate is 67.0 $\mu$s long and centered at 1095.2 $\mu$s. For each stack, the 252 transients are sign-corrected and a motion-noise filter is applied (Auken et al., 2019). The filter output contains 204 transients from which we compute mean values and standard errors (Appendix A).

A representative high moment sounding curve from the Gedved survey containing 81 sub-gates is plotted in Fig. 3. We

observe a smooth decay in the TEM signal until ∼ 100 $\mu$s. From 100 $\mu$s and onwards oscillations are seen in the sounding curve. The inset in the figure shows the TEM signal from 100 $\mu$s to 200 $\mu$s and the associated standard errors. Importantly, the errors are significantly smaller than the observed oscillations in data and hence the oscillations cannot be interpreted as the result of random noise. We ascribe the oscillations to VLF noise, which in this context can be considered a coherent noise source, which is not efficiently suppressed by stacking of transients. Our interpretation is supported by an analysis of the peak-

to-peak separations of the oscillations. The separations correspond to frequencies in the 20-25 kHz range in agreement with VLF radio stations observed in Denmark (Rasmussen et al., 2018b).

Next, for each transient, we assemble linear combinations of the short sub-gates into longer, exponentially widening gates either boxcar or semi-tapered with a target of ten gates per decade. In the case of boxcar shaped gates, the gates from 100 $\mu$s



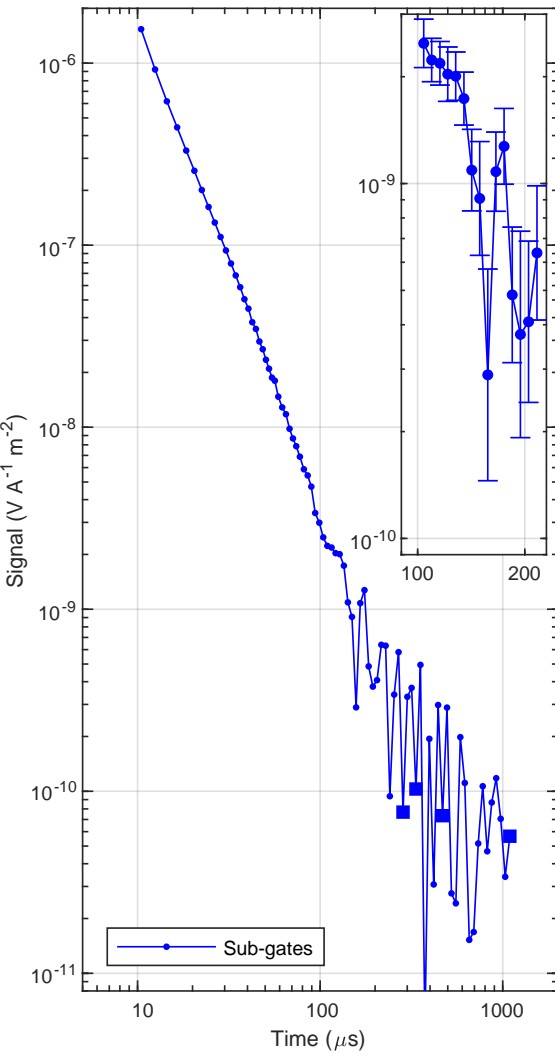

**Figure 3.** Sounding curve from the Gender survey consisting of 81 boxcar-shaped sub-gates. For negative valued sub-gates, the absolute value is plotted and indicated with a square marker. From approximately 100 $\mu$s and onwards the data is dominated by VLF noise and oscillates. The inset shows values and standard errors for sub-gates between 100 $\mu$s and 200 $\mu$s.

and onwards are composed of five or six sub-gates controlled by the exact distribution of sub-gates. For semi-tapered gates,

the flat top contains the same five or six sub-gates and the tapers overlap with the previous and next boxcar gate. Each taper contains five or six sub-gates with values matching a smooth half-cosine taper at the sub-gate centers. The only exceptions are the last two gates, where the finite number of available sub-gates enforces a non-cosine off-tapering. For the last gate, only three sub-gates are available for the boxcar gate and the flat-top part of the semi-tapered gate. The subsequent processing of assembled data is similar to above. Data are sign-corrected, motion-noise filtered and stacked.



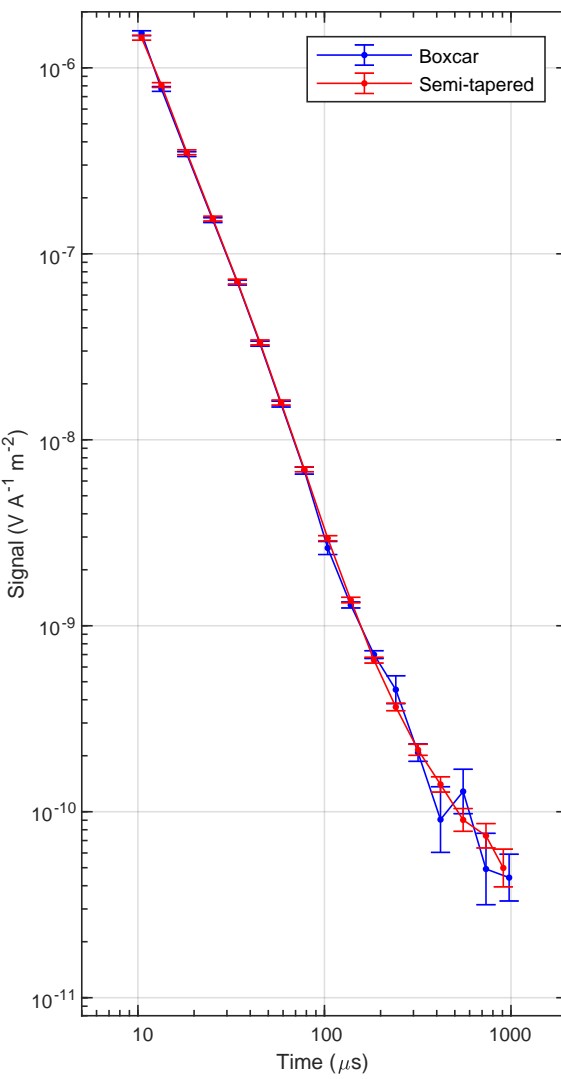

**Figure 4.** Sounding curves, boxcar and semi-tapered, from the Gedved survey. Each sounding curve contains 17 gates. The boxcar and semi-tapered gates are constructed from the sub-gates shown in Fig. 3.

In Fig. 4 we plot an example of a boxcar and a semi-tapered sounding curve, with gate 4-20 in each. The first three gates are distorted by transmitter turn-off effects and therefore culled. The two sounding curves are very nearly identical until ∼200 $\mu$s. After this time, the semi-tapered curve is continuous with a smooth decay, whereas the boxcar sounding curve still shows oscillations. Further, the error bars on the boxcar gate values are larger than for the semi-tapered gates. Compared to the 81-gate sounding curve in Fig. 3, we see that the increased width of the boxcar gates extends the range of usable data but is





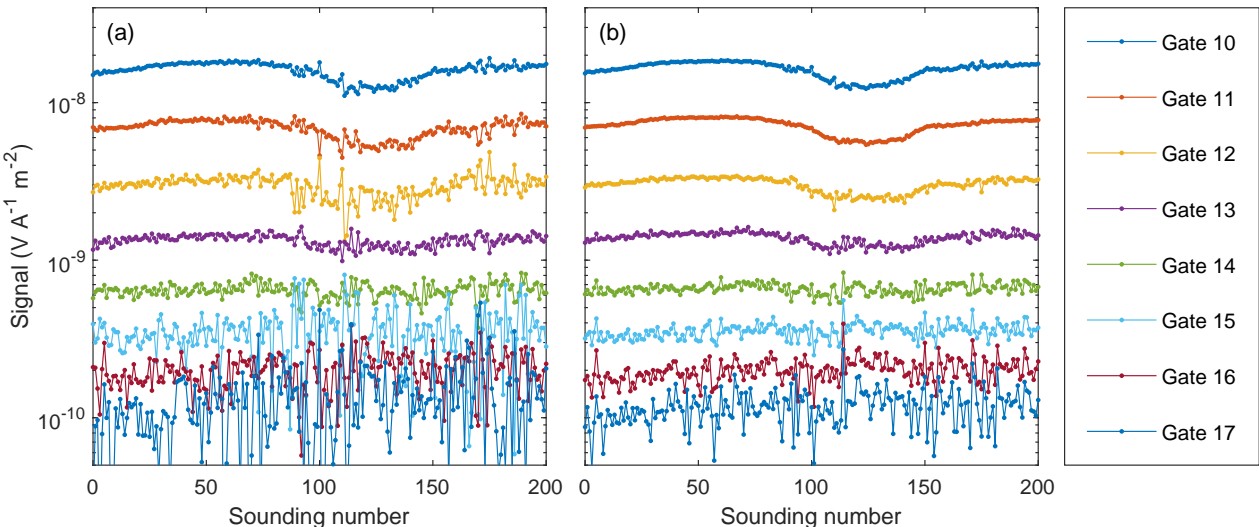

**Figure 5.** Gate values as a function of sounding index acquired with the tTEM system moving (a) boxcar gates, (b) semi-tapered gates, see Table 1 for gate details. The 200 soundings corresponds to approximately 200 m of driving. The noise using semi-tapered gates is visibly lower than for boxcar gates.

incapable of sufficient reduction after $\sim$200 $\mu$s. In contrast, the semi-tapering is significantly better to suppress VLF noise. The sounding curve is much smoother and follows the normal decay of a transient.

Figure 5 shows 200 subsequent soundings from the Gedved survey with sub-gates gated into either standard-length boxcar gates, Fig. 5(a) or semi-tapered gates, Fig. 5(b). Gates 10-17 with gate centers from 59.2 $\mu$s to 978.1 $\mu$s and gate widths from 14.6 $\mu$s to 394.9 $\mu$s are plotted (Table 1). It takes a little less than 2 minutes to record the 200 soundings during which the tTEM system moves approximately 200 m. The plots show a clear suppression of noise with the semi-tapered gating strategy as compared to the boxcar gating strategy for all gates. Earlier gates are visibly very similar for the two gating strategies and therefore not plotted. Gates 18-20 also show improvement, but these gates are not shown, as the noise is too large in these late gates for an easy visual comparison on the same plot. The noise is less pronounced in the first about 75 soundings indicating a strong nonstationary noise field.

We quantify the improvement in noise reduction by defining an improvement factor, $\gamma$, as the standard error of data measured with boxcar gates divided by the standard error of data measured with semi-tapered gates, i.e.,

$$\gamma = \frac{S_{boxcar}}{S_{semi-tapered}}. \tag{1}$$

The improvement factor is above one if the standard error of data measured with semi-tapered gates is less than the standard error of data measured with rectangular gates, i.e., if the noise reduction is improved with semi-tapered gates and vice versa.

In Fig. 6 we plot histograms of the improvement factors for gate 9-20 measured on 1825 soundings from the Gedved survey. On top of the histograms, mean value and standard error of the improvement are indicated with red lines. Starting from gate





**Table 1.** Gate center times, gate width, improvement factors, $\gamma$ and standard deviation for gates 9 to 20. Gate widths are measured as full width at half maximum. Boxcar and semi-tapered gates have the same center time except for the last gate (20), where the table gives the center time of the boxcar gate. The corresponding semi-tapered gate center time is 910.4 $\mu$s.

| Gate no. | Gate center ($\mu$s) | Boxcar gate width ($\mu$s) | Semi-tapered gate width ($\mu$s) | Impr. factor $\gamma$ |
|---|---|---|---|---|
| 9 | 45.8 | 11.6 | 23.4 | 0.99±0.02 |
| 10 | 59.2 | 14.6 | 29.5 | 1.02±0.05 |
| 11 | 78.7 | 24.6 | 49.8 | 1.07±0.12 |
| 12 | 104.8 | 26.6 | 53.7 | 1.26±0.32 |
| 13 | 139.1 | 42.6 | 86.2 | 1.04±0.08 |
| 14 | 185.1 | 48.6 | 98.1 | 1.12±0.12 |
| 15 | 242.0 | 65.6 | 132.4 | 2.08±1.01 |
| 16 | 318.2 | 87.6 | 176.9 | 1.31±0.27 |
| 17 | 419.7 | 114.6 | 235.5 | 2.22±1.06 |
| 18 | 554.8 | 155.6 | 314.3 | 1.88±0.68 |
| 19 | 735.3 | 208.6 | 421.4 | 1.98±0.88 |
| 20 | 978.1 | 281.6 | 394.9 | 1.18±0.16 |

15, centered at 242 $\mu$s, we see significant improvement factors, with the mean close to or exceeding two in four gates. In gate 16, the mean improvement is 1.31, which is remarkable low compared to the improvement in neighboring gates. This is likely caused by a specific high-amplitude VLF radio transmitter being well suppressed by a zero in the boxcar frequency response.

For the last gate, 20, we also see a low improvement factor of 1.19. This low value is due to the truncated semi-tapered gate having a non-optimum frequency response close to the boxcar frequency response. The results are summarized in Table 1. We observed no improvement for the early gates (before gate 9). This result is anticipated as the main lobe using either gating strategy exceeds well beyond the 3-30 kHz VLF band.

    If the improvement factors are analyzed as functions of the gate values, on average, negative correlations are seen. This

result is intuitively clear, as high gate values generally correspond to large TEM signals where the impact from VLF noise is smaller and hence the potential improvement is smaller.

    We assess the impact of the semi-tapering in the resistivity model space by smooth inversion of the two differently gated data sets. During processing of data, all gates with a standard error exceeding 10 % of the gate value are culled together with any later gates in the sounding. Similarly, all negative data values are culled. As part of the inversion process we also compute

the depth of investigation (DOI) for each sounding with the approach developed by Christiansen and Auken (2012). In Fig. 7,





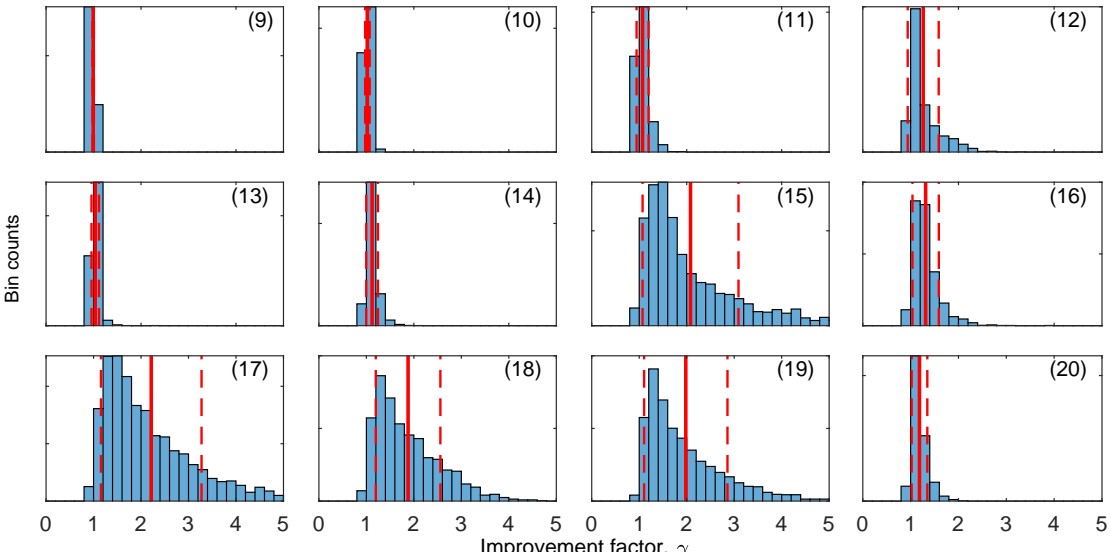

**Figure 6.** Histogram of improvement factors for gates 9-20 presented as bin counts. The bin count axis is individually scaled for each histogram to make the distribution visible. The full red line shows the mean value of improvement factor, the dashed red lines show the standard deviation.

we present a scatter plot of DOI's for the 1825 soundings. On average, in this case we find that semi-tapered gating increases the DOI by 15 m or about 20 %.

We also compute the average logarithmic data residual, $\phi$, of the inverted models using

$$\phi = \sqrt{\frac{1}{N_g} \sum_{k=1}^{N_g} \left( \frac{\ln(X_k^{observed}) - \ln(X_k^{predicted})}{\ln(1 + S_k/X_k^{observed})} \right)^2} \qquad (2)$$

where $N_g$ is the number of gates and $X_k$ denotes the $k$'th observed or predicted gate value with standard error $S_k$ (Appendix A). For boxcar gating the average data residual is 1.38, whereas this value is only 1.09 using semi-tapered gating. Our interpretation of this difference is that the standard errors on the boxcar gate values are not truly representative of the noise. This occurs as the standard error reflects only the random noise in the data, but coherent VLF noise is not properly accounted for. Using semi-tapered gating, the coherent VLF noise is suppressed and the standard errors much better reflects the actual noise in the

data.

## 4 Discussion

The semi-tapered gates that we employ are based on augmenting boxcar gates with tapering on both sides of the original boxcar gate. The improvement in noise suppression is therefore a combination of two properties. First, the augmenting with tapers increases the width of the semi-tapered gates as seen in Table 1. This leads to a narrower main lobe and hence by itself





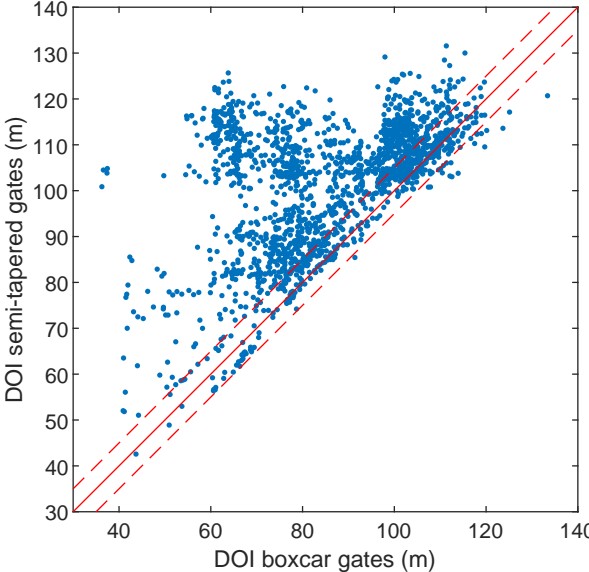

**Figure 7.** Scatter plot of depth of investigation (DOI) using either boxcar or semi-tapered gating. The full red line indicate equality of DOI, while the two red dashed lines indicate a DOI difference of 5 m. The average increase in DOI using semi-tapered gating is 15 m.

improved suppression of low-frequency noise. Second, the tapering gives a magnitude response with suppressed side lobes and hence also an improved noise rejection.

In the data from the Gedved survey, sub-gates are visibly influenced by VLF noise from around 100 $\mu$s and onwards, and our analysis has focused on this part of the data. It is also possible to use the semi-tapering approach with earlier gates. For this data set, the sub-gates recorded before $\sim$100 $\mu$s generally have high SNR and the need for additional noise reduction is absent.

In our approach where we use additional sub-gates for tapering, the semi-tapering increases the width of any given gate. In turn, this reduces the main lobe width, which can influence the desired TEM signal, especially at early times where the TEM signal is high bandwidth. However, for surveys conducted on high-resistivity sub-surfaces, the TEM signal will disappear into noise at earlier times and starting semi-tapering of sub-gates earlier than 100 $\mu$s can be beneficial.

In noise scenarios, where the majority of noise stems from a single powerful narrowband radio transmitter, it can be beneficial

still to employ boxcar gates or to use more advanced semi-tapering schemes. Figure 2(b) shows that boxcar gates have zeros in the magnitude response. Hence, by choosing the width of a boxcar gate appropriately, a zero can be placed directly at the frequency of the radio transmitter resulting in a very high noise rejection. Likewise, more advanced gating strategies are also possible. By tuning the width of boxcar sub-gates as well as the width and tapering of combined gates, a set of semi-tapered gates can be optimized for the exact noise conditions at the site where a survey is carried out. As a simple example of the

potential of this approach, we carried out an experiment on the Gedved data where a new gate 16 was assembled by shifting all sub-gates one step earlier. This shifted the gate center from 318.2 $\mu$s to 300.9 $\mu$s, while the gate width was reduced from 87.2 $\mu$s to 82.6 $\mu$s (boxcar) and from 179.9 $\mu$s to 166.8 $\mu$s (semi-tapered). The outcome of the shifting operation is a significant





increase in the improvement factor from 1.31±0.27 to 1.63±0.60 for gate 16. For the other gates, some small increases as well as some small decreases in the improvement factor was observed.

For a given survey, the actual improvement in SNR and in turn improved depth of investigation can vary significantly depending on the resistivity structures and the local noise environment. In particular, we have observed that rough driving conditions can induce motion noise that post-filtering still exceeds the contributions from VLF noise. In such a case where TEM measurements are dominated by motion-induced noise, the performance of semi-tapered and boxcar gating strategies are on par. The data set that has been used for this work stems from an area with low to medium sub-surface resistivity, which give

rise to large amplitude TEM signals. In surveys performed over high-resistivity sub-surfaces, the TEM signal, particularly in the late gates, will be significantly lower and even better results than presented here can be expected in many cases.

One attractive feature of semi-tapering is its ability to suppress coherent VLF noise, which cannot be achieved by stacking of data. The VLF noise suppression leads to estimates of error bars that are more truly representative of the actual noise in data. Further, the suppression of VLF noise reduces systematic errors in data, as can be seen in Fig. 4 where the semi-tapered gating

results in a smoother decay than the boxcar gating. In turn, this leads to resistivity models with better determined resistivity parameters and larger depth of investigation.

This study has shown that boxcar gates can efficiently be turned into semi-tapered gates with close to the same noise suppression capability as fully sampled transients. From an instrument design perspective this is quite important because modern systems can use A/D converters sampling at 10 MHz or more. Storing this amount of data is not possible. Decimating

the data stream into boxcar gates can reduce the storage to maybe 100–200 gates per transient, which is unproblematic and it allows for re-gating during data processing.

## 5   Conclusions

We have explored a new gating strategy applicable for TEM systems equipped with analog boxcar integrators. The gating strategy is based on acquiring many short boxcar gates, which are then combined into a set of overlapping semi-tapered gates.

The semi-tapered gates were experimentally shown to have better noise suppression than the standard boxcar gates. The noise suppression was quantified by comparing the standard error of data acquired with each approach and through the depth of investigation. We found that for this specific data set, the standard error was decreased by a factor between 1.04 and 2.21 and the depth of investigation was increased by 20 % using the semi-tapered gates. The increase in SNR is achieved at no expense, except a reprogramming of the gating scheme.

The reduced standard errors on data translates directly to an increased SNR. If the same improvement in SNR were to be obtained by increasing the signal through an increased transmitter current, the current would have to be increased by the same factor (in this case a factor of 2), which is a significant engineering challenge.



## Appendix A: Calculation of uncertainty on gate values

The tTEM system repeats the measurement of each transient $N_t$ times and each transient contains $N_{sg}$ sub-gates. We denote the
value of the $i$'th transient and $j$'th sub-gate as $\tilde{x}_{ij}$ with $i = 1, \ldots, N_t$ and $j = 1, \ldots, N_{sg}$. The measured data are sign-corrected
and motion-noise filtered, then processed as follows. Note that the motion-noise filter reduces $N_t$ by the length of the employed
filter as data with transient filter distortion are discarded. For each transient, the data are first regated into $N_g$ longer boxcar or
semi-tapered gates using

$$x_{ik} = \sum_{j=1}^{N_{sg}} \tilde{x}_{ij} w_{jk} \tag{A1}$$

where $k = 1, \ldots, N_g$. nd $w_{jk}$ is the weight of the $j$'th sub-gate in the $k$'th gate. The weighting factors are normalized to a unit
magnitude response at 0 Hz. Second, the mean (stacked) value of the $k$'th gate, $X_k$, is computed as

$$X_k = \frac{1}{N_t} \sum_{i=1}^{N_t} x_{ik}. \tag{A2}$$

Next, the mean square deviation is computed with

$$s_k^2 = \frac{1}{N_t - 1} \sum_{i=1}^{N_t} (x_{ik} - X_k)^2. \tag{A3}$$

Finally, the standard error for the stacked measurement is estimated as

$$S_k = \frac{s_k}{\sqrt{N_t}}. \tag{A4}$$

The values of stacked data are reported along with their standard error as $X_k \pm S_k$ (Barford, 1990).

*Author contributions.* The idea was conceived by EA and JJL. Data collection and data analysis was performed by SSP, NF and JJL. Analysis
and interpretation was done by SSP, NF, EA, and JJL. The manuscript was written by SSP and JJL.

*Competing interests.* The authors declare that they no conflict of interests.

*Acknowledgements.* Pradip K. Maurya is kindly acknowledged for help with tTEM field measurements and data processing. The tTEM data
was collected during TOPSOIL, an Interreg project supported by the North Sea Programme of the European Regional Development Fund of
the European Union. This research is supported by Innovation Fund Denmark through the MapField project.





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
