# Peer review of "Suppression of very low frequency radio noise in transient electromagnetic data with semi-tapered gates"

_Geoscientific Instrumentation, Methods and Data Systems, 2020_

## Referee Comment (RC1) · James Macnae (Referee) · 24 Jan 2021

In a TEM system, the gates are stacked with alternating signs, to ccount for the alternating transmitter waveform. In this case, the frequency domain response of the windowing scheme comes from taking the Fourier transform of a series of alternating sign gates (numerically an fft of a pair of alternated gates). It appears to me that Figure 2 is the transform of a gate (or maybe gates) of positive sign, and thus is unrepresentative of the gating scheme used in tTEM and other geophysical systems. May I suggest the authors look at this issue of repetitive sampling?

While not published in the literature, it is fairly well known that most airborne (e.g.

[Figure]

Geotech VTEM, CGG Geotem) and some ground geophysical systems (e.g. Smartem) use tapered stacking, which is derived digitally from streamed time series, essentially a version of the semi-tapered gates described here. This paper however, does describe this process and would be a very useful reference for future work.

James Macnae
* * *

---

## Author Comment (AC1) · 27 Jan 2021

Dear James

Thank you very much for your review and comments.

Regarding figure 2. You are completely right that this figure only shows the transform of a single gate. The effect of repetitive sampling is to give a comb-like sampling of the single gate frequency response. This comb-like sampling proceeds identically, irrespective of the choice of the single gate shape. Therefore the lessons learned from figure 2 still apply. In the revision of the manuscript we will be more explicit about what

figure 2 shows and clarify the down-stream effects of repetitive sampling (most likely by adding an appendix with the mathematics of this).

Kind regards

Jakob

---

## Referee Comment (RC2) · Marco Antonio Couto Junior (Referee) · 4 Feb 2021

Dear authors,

I think the results presented in the paper are really good. I could not notice almost any text issues in the manuscript, just one out of nonstandard citation in line 99: where you put Harris (1978) should be (Harris, 1978).

For future works, or maybe as an addition to this one, some results related to very noisy environments could be integrated. For example, for airborne TEM data in highly mountainous terrain and mining districts, where the mining activity noise could be significant,

even prohibitive in case to identify IP signals related to economic interest anomalies, for example. For the IP case, how to deal with the negative decays, as you culled them off in your procedure? Also, these IP signals might have standard error above the 10% criteria. How your strategy will behave if this threshold needs to be increased?

In any case, I think the paper can be accepted as it is, with the only citation correction I have mentioned above.

Congratultions for this work. I liked it a lot.

---

## Author Comment (AC2) · 6 Feb 2021

Dear Marco

Thank you for the positive comments.

Regarding your comment on IP effects. The culling of negative data and removal of the worst 10% of data is done during the inversion of data, not during the semi-tapering. More data can be kept in the inversion if desired.

In the scenarios you point out, semi-tapered gating is not necessarily the optimum approach. Tapered (and semi-tapered) gating corresponds to filtering of data and hence

suppress specific frequency bands. Most likely, a better option is to use a full-sampled TEM receiver system where the VLF noise can be modelled and subtracted from the data (Macnae 2015, Rasmussen et al. 2018b). Ideally, this will leave the TEM signal unaffected and gating schemes optimized for enhancing IP effects can be used. Specific suppression of mining noise would demand a detailed study and understanding of this particular noise source, which is beyond the scope of this paper.

Kind regards

Jakob

---

## Author Response (AR1)

Geoscientific Instrumentation, Methods and Data Systems
Associate Editor Lev Eppelbaum

**GI-2020-49 revised manuscript**

Dear Lev Eppelbaum / GI editors

We hereby submit the revised version of our manuscript, "Suppression of very low frequency noise in transient electromagnetic data with semi-tapered gates". We would like to thank the two reviewers for their acknowledging comments and their assessment of our work.

In the revised version, it is now explicitly stated in section 2.1, that figure 2 and the accompanying discussion refers to a single gate. Section 2.1 is expanded with a final paragraph commenting on the effects from repetitive sampling. The mathematics behind repetitive sampling is derived in a new appendix A at the end of the manuscript. These revisions are marked with blue in the annotated manuscript.

On behalf of the authors

Jakob Juul Larsen
Associate Professor

**Sign. Processing & Mach. Learning**

**Jakob Juul Larsen**
Associate Professor

Date: 24 February 2021

Direct Tel.: +45 4189 3273
E-mail: jjl@ece.au.dk
Web: au.dk/en/jjl@ece

Sender's CVR no.:
31119103

Page 1/1

[Figure]

**Signal Processing and Machine learning**
Aarhus University
Finlandsgade 22
DK-8200 Aarhus N
Denmark

Tel.: +45 8715 0000
E-mail: ece@au.dk
Web: ece.au.dk/en